# Reconstruction of ovine axonal cytoarchitecture enables more accurate models of brain biomechanics

Andrea Bernardini [1,5 ✉], Marco Trovatelli[2,5], Michał M. Kłosowski[3], Matteo Pederzani[4], Davide Danilo Zani[2], Stefano Brizzola[2], Alexandra Porter[3], Ferdinando Rodriguez y Baena[1] & Daniele Dini [1 ✉]

There is an increased need and focus to understand how local brain microstructure affects the transport of drug molecules directly administered to the brain tissue, for example in convection-enhanced delivery procedures. This study reports a systematic attempt to characterize the cytoarchitecture of commissural, long association and projection fibres, namely the corpus callosum, the fornix and the corona radiata, with the specific aim to map different regions of the tissue and provide essential information for the development of accurate models of brain biomechanics. Ovine samples are imaged using scanning electron microscopy combined with focused ion beam milling to generate 3D volume reconstructions of the tissue at subcellular spatial resolution. Focus is placed on the characteristic cytological feature of the white matter: the axons and their alignment in the tissue. For each tract, a 3D reconstruction of relatively large volumes, including a significant number of axons, is performed and outer axonal ellipticity, outer axonal cross-sectional area and their relative perimeter are measured. The study of well-resolved microstructural features provides useful insight into the fibrous organization of the tissue, whose micromechanical behaviour is that of a composite material presenting elliptical tortuous tubular axonal structures embedded in the extra-cellular matrix. Drug flow can be captured through microstructurally-based models using 3D volumes, either reconstructed directly from images or generated in silico using parameters extracted from the database of images, leading to a workflow to enable physically-accurate simulations of drug delivery to the targeted tissue.

---

[1] Department of Mechanical Engineering, Imperial College London, London SW7 2AZ, UK. [2] Faculty of Veterinary Medicine, Università degli Studi di Milano Statale, 26900 Lodi, Italy. [3] Department of Materials, Imperial College London, London SW7 2AZ, UK. [4] Department of Electronics, Information and Bioengineering, Politecnico di Milano, 20133 Milan, Italy. [5]These authors contributed equally: Andrea Bernardini, Marco Trovatelli. ✉email: a.bernardini16@imperial.ac.uk; d.dini@imperial.ac.uk

Brain is a complex organ where a multitude of cells cooperate toward its structural integrity and functionality. Neurons, astrocytes, oligodendrocytes, and microglia are among its main cellular bodies that can be broadly classified as neurons and non-neurons, namely neuroglia[1]. The cerebral White Matter (WM) is where electric signals from neurons travel through their fibrous structures: the axons. These are neuronal protrusions supported by neuroglia and wrapped in myelin, an insulating lipidic layer. This conformation gives WM a specific biomechanical behaviour, characterized by the directionality and geometry of these fibrous structures[2,3].

Axons have filamentous components (i.e. neurofilaments, microtubules and microfilaments) that lead to a stiffness[4] capable of a load bearing function. Conversely, neuroglia show a very soft nature and act as a soft embedding matrix for the axons[5]. Because of its particular nature, there has been growing interest in topological and mechanical characterisation[6–8] and modelling[9–11] of WM in different studies like tumour proliferation[12,13], traumatic brain injuries[14,15] and Convection Enhanced Delivery (CED) research[16–21]. Axons are important mechanical features of WM; in fact, WM stiffness is highly related to their presence. The axonal cytoskeletal elements such as microtubules contribute greatly to the axon mechanical response[4] and its relative contribution to the overall tissue stiffness. Also, it has been proved that the content of the axonal myelin wrapping the axons linearly increased the stiffness of the WM tissue[22] and makes them impermeable due to this hydrophobic lipid[23–28]. Furthermore, CED studies and clinical investigations of the cerebral oedema propagation show a clear influence of the directionality and organisation of axonal fibres on fluid flow and diffusion within the tissue[13,29–32].

However, this mounting interest in the biomedical engineering community for the WM structure has not been met and matched by ultrastructural histological studies, which focus on characterising the health of the tissue's rather than the relationship between 3D structure and mechanics at this length scale. In fact, their main aim is still to image cytological structures and substructures[33–35], with a final scope of describing the structure of cells, especially to study pathologies and their nanoscale effects. These hyper-localized studies rely on images from high-resolution microscopy techniques, i.e. serial block-face Scanning Electron Microscopy (SEM)[36–38], Focused Ion Beam Scanning Electron Microscopy (FIB-SEM)[39,40], or Transmission Electron Microscopy (TEM)[41,42]. At the organ level, other research[27,43] employs different technologies, i.e. Magnetic Resonance Imaging (MRI), to investigate the fibrous macrostructure of the whole of WM for neuroanatomical purposes. However, their achievable resolutions of 2–5 μm miss a large percentage of smaller axons that end up undetected[44]. Many advances have been made in this area over the past two decades; there have been noticeable improvements in both high-resolution microscopy techniques and the ability to segment and reconstruct brain white matter microstructure with increased accuracy and varied degree of automation[45–49]. Furthermore, significant developments have been made in the use of non-destructive techniques employing e.g. polarised light imaging and optical coherence tomography[50–54] to study the fibres orientation and brain activity in live tissues, which have culminated in the concrete effort to deliver protocols for correlative super-resolution imaging of samples[55] and, at the same time, to attempt the bridging between scales using a combination of diffusion MRI, 3D synchrotron X-ray nano-holotomography images and 3D high resolution microscopy[56].

Here, we aim to provide fundamental data and models to fill the gap between histology and biomechanics (Fig. 1) by creating the first database of histological measurements specifically aimed at microstructurally-based mechanical modelling of the WM across different regions of the brain. Common ground was sought

between the two imaging approaches by applying a serial FIB-SEM imaging technique (resolution of 0.020 μm/pixel) to visualise, in 3D, three different areas of the WM to characterize different WM areas and to shed a first light on any possible regional differences. This technique combines the high spatial resolution (~3 nm) of SEM imaging with the ability to section through samples to generate 3D volumes and is uniquely effective for 3D reconstruction of brain tissue at high resolution. In this study we resort to manual segmentation, as discussed in the *Methods* section of the paper. While this is laborious, it has enabled us to carefully identify and reconstruct the myelinated structures of interest. Alternative advanced techniques have recently emerged that allow semi-automatic and automatic segmentation of WM microstructure based on the availability of large datasets[45,46,56,57], as well as tools to aid all types of segmentations[49]. These investigations offer interesting pipelines for segmentation and quantification of white matter samples and will be further discussed in the Discussion section, also highlighting how the reconstructed set of 3D samples generated in the present study could be used for

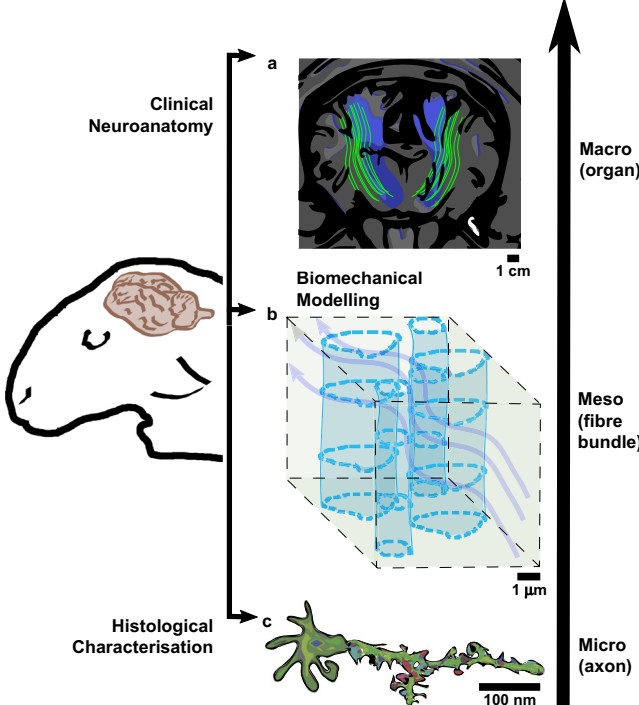

**Fig. 1 Different length scales of interest exist within the scientific research community. a** Typical clinical images, such as the tractography[107], give a broader field of view to clinicians. Medical research is interested in the macro-organization of the WM as via these images the diagnosis, surgical planning and the overall assessment of the brain can be studied and achieved. **b** At the intermediate length scales, microstructural information can be obtained to reconstruct axonal structures, aiming at a final accurate geometrical characterization of a Representative Volume Element (RVE), which can be used to shed light on fundamental aspects of tissue biomechanics, e.g. mechanical[9] and fluid dynamics[108], or material mimicking studies[109]. **c** At the smallest scales 3D reconstructions of individual axons[103] can be performed, which give a detailed representation of how the axons appear in grey matter: branching of the axons, synaptic buttons and dendritic spines can be easily distinguished, aiding the histological understanding and description up to the finest detail of neuronal bodies. However, these two categories of data either miss the simplification needed by the bioengineering community or lack the length scale of investigation that are fundamental to relate the stiffness of the tissue to its structure at the length scale of the myelinated axons.

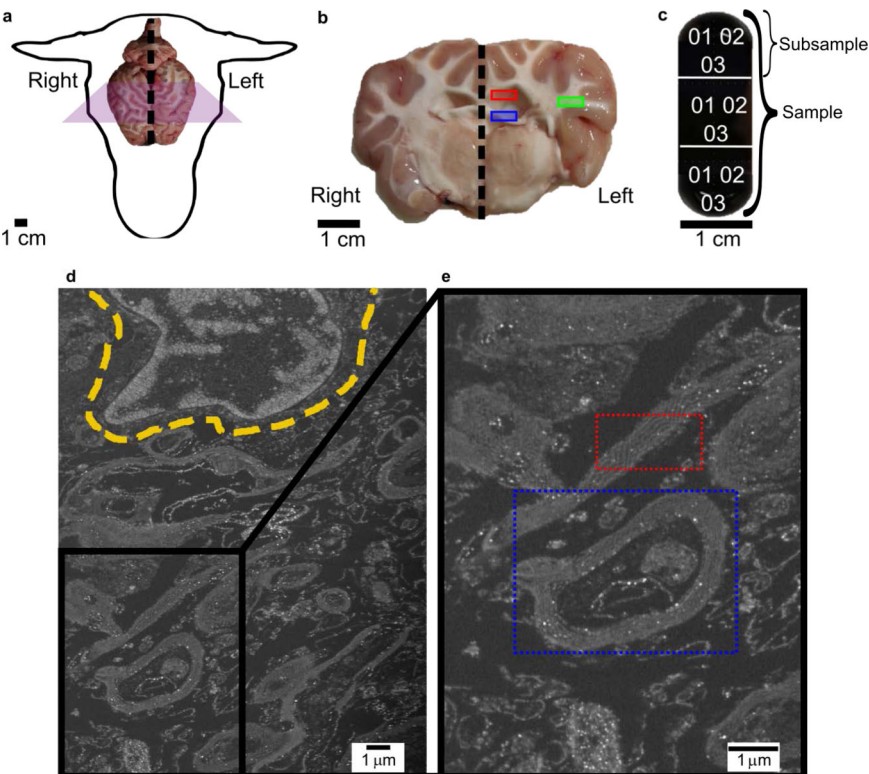

**Fig. 2 Sampling from ovine cerebrum and acquired 2D FIB-SEM images. a** Each brain was excised and immediately sampled. The samples have always been taken from a specific part of the brain: the left hemisphere. **b** The samples come from different regions of the brain, samples from the Corpus Callosum (red box) being a Commissural Pathway, have fibres that connect the two different hemispheres. Projection Pathways like the Corona Radiata in the green box, connect the cortex to subcortical structures. Finally, the long association pathway, the Fornix in the blue box, also connects regions that belong to the same hemisphere. **c** After fixation, dehydration and embedding, the tissue is stored into capsule-like samples that follow a three-way subdivision for the imaging process that consisted in imaging three random locations in each of the subsamples for the 2D analysis and one extra location for the 3D analysis. **d** The stained tissue imaged by SEM shows the major cellular components of WM. The accessory cells (neuroglia), like the one in the yellow-dashed line, appear as blobs with ramified protrusions reaching out in the interstitial spaces. **e** The axons appear as white ringed structures (dashed blue box) as they are wrapped in their myelin sheaths and at a further magnification it can be seen where the membranes of specialized accessory cells, the oligodendrocytes, come appose and appear as white and dark bands packed together in a layer (dashed red box).

validation purposes and to further improve techniques adopted for semi-automatic and automatic segmentation in the presence of sparse datasets.

Three different areas, in three different *ovine* brains have been sampled: *Corpus Callosum* (CC), *Corona Radiata* (CR) and *Fornix* (FO). Ovine samples were chosen as part of a larger study, ultimately aimed at performing in vivo CED trials on animal samples, since sheep are increasingly recognized in several fields of translational neuroscience[58]. The use of three brains from different subjects enables us to perform a rigorous study of the characteristics of the key microstructural parameters for different regions of the brain in analogy of studies performed for examples on human brain samples[44]. This was deemed to be a good compromise between the (already significant) time required to acquire the images across different regions of the individual animals and the need to investigate intra-subject variations. The initial slicing of the tissue (Fig. 2) was performed so that different fibre tracts could be easily accessed, and samples could be consistently excised from the WM in the left hemisphere of each individual brain. After fixation in formaldehyde and staining in Osmium Tetroxide ($OsO_4$), the resin-embedded samples were imaged *via* FIB-SEM. Analyses of 2D images were used in the first instance to estimate the homogeneous myelination content of lipids in all if the different fibre tracts. Additionally, for each tract, a 3D reconstruction of the axons in the imaged volumes (average dimensions of $\sim15 \times 15 \times 15 \ \mu m^3$) has been performed and axonal measurements taken, namely: outer axonal ellipticity, outer

axonal cross-sectional area, outer best-fit diameter and axonal tortuosity.

This study provides a systematic approach to the characterization of tissue for engineering purposes, and it gives a useful insight into the 3D fibrous structure and organisation of the WM. From a biomedical engineering perspective brain WM is a composite material comprised of a soft embedding matrix with inclusions of relatively straight, elliptical tubular fibres. With the aim to map the cytoarchitecture of a broad range of WM regions, we provide a detailed analysis of the distribution of axonal area and diameter at the cross-section based on the full 3D ovine tissue reconstructions of three regions of the brain. These reconstructions and the geometrical parameters obtained from their analysis can be used directly to perform deterministic studies of WM permeability and to determine tissue deformations and mechanical response[59–62]. Finally, to show the rapid and easy implementation of such data into modelling techniques, three periodical Representative Volume Element (RVE) have been reconstructed, one per each area studied, by using the extrapolated probability distributions of the measured data.

## Results

**Image acquisition and post processing.** A schematic description of the sampling areas in an *ovine* brain is shown in Fig. 2a, b. From each of the main samples excised from the different fibre tracts (Fig. 2b), three subsamples were studied by imaging

different 2D areas at random locations (Fig. 2c). This enables the investigation of the homogeneity in the sample composition. Finally, for each sample corresponding to each fibre tract, a full 3D scan from one of the subsamples has been acquired per each subject with an average volume of about $15 \times 15 \times 15\ \mu m^3$. With a final resolution of $0.020\ \mu m$ per pixel, the total acquisition time resulted in ~8 h per each 3D volume using the protocol described in *Methods*. It is important to note that this acquisition protocol has been optimised to enable relatively large volumes to be scanned while preserving the resolution required for the accurate reconstruction of axonal structures. The volumes analysed here, which have been selected according to the recommendation provided when studying large-volume 3D imaging using FIB-SEM[40], are in line with the tissue volumes generally targeted by high-resolution EM reported in the recent literature[46]; larger samples sizes with resolution acceptable to identify and reconstruct axonal features can currently be achieved for example with 3D synchrotron X-ray nano-holotomography[56].

The 2D areas have been acquired at random locations and with different plane of cuts relative to the main directionality of the fibre tracts resulting in images like the one in Fig. 2d, where different features can be distinguished (Fig. 2e). Figure 3a–c show representative SEM images from corpus callosum, corona radiata and fornix respectively. To assess the homogeneity of the sample, which is one of the aims of this study, and therefore the statistical relevance of the location randomly chosen to perform 3D tomography, a qualitative analysis *via* pixel counting of the binarised images was carried out *via* MATLAB scripts[63]. The noise was initially reduced in a two-step procedure. The first step involves a reduction of the "salt and pepper" noise *via* a median filtering where the median values per each pixel is chosen in the $3 \times 3$ neighbourhood with symmetrical padding at the boundaries. This reduction of the grey values scatter around the mode reduces the amount of spurious noisy pixels. Ideally, myelin would be imaged as "continuous rings" so isolated white pixels are not representative of the myelin content. Therefore, the remaining noisy, spurious and isolated pixels have been removed *via* additional morphological operations such as opening (Fig. 3d). It should also be noted here that some artefacts, such as limited degree of myelin delamination, are observed in some of the SEM images. This, which is not uncommon in EM images of brain matter[44,48], cannot be associated with tissue damage induced by pathological conditions here and is likely to be linked to alteration induced by the fixation and/or post-mortem time of excision of the brain samples[48,64] (the latter being less of a concern in this study—see Discussion).

The different acquisition positions can lead to the areas of interest being differently exposed to the beam causing uneven background illumination. This hinders the ability to directly compare the grey levels distribution and the distribution of the material in the sample. Therefore, as a second step, a homogenization of the background illumination has been performed. *Via* morphological opening the objects in the foreground (the axonal matter) have been removed resulting in solely the unevenly illuminated background (Fig. 3e). Subsequent subtraction of this background from the original image reduces the irregularity.

After reducing the noise, image thresholding operations have been carried out to represent the images in binary form. The MATLAB function *adaptthresh*[65], that locally computes threshold values per each pixel, has been used to remove any persistence of the heterogeneously illuminated background. This preserves significant contrast without taking into account gradient changes and it counterbalances the remaining uneven exposure of the background.

After final binarisation (Fig. 3f), a calculation of relative frequencies of white pixels, defined here as the ratio between white pixels and the overall number of pixels, has been carried out in each image to estimate the myelin content.

**Composition of cytostructure**. A comparison has been carried out to assess compositional homogeneity of the samples and to perform a comparison between association (corpus callosum), commissural (corona radiata) and projectional (fornix) fibres. The average pixel counts have been computed from each area (area 1, area 2 and area 3) and have been compared against each of the three subsampling zones: 01, 02, and 03 (Fig. 3). The average number of white pixels imaged from each subsampling area shows a percentage of around 40% on the total pixel count in all subjects. In each sample, the differences between subsampling areas (01 02 03) are minimal. The average values are always within the biggest standard deviation of the three subsamples (Fig. 3).

Turning now to the analysis of the 3D reconstructed samples, each individual axon was manually segmented for all samples identified from 3D tomographic analysis (Fig. 4a), as discussed in the *Methods* section. After calculating a centreline for each axon *via* the built-in MIMICS function[66], the quantities of interest have been measured on the cross-sectional planes perpendicular to the centrelines. This allows an exact measurement of the relevant geometrical properties regardless of the position of the acquisition plane relative to the instances (Fig. 4b).

The calculations of cross-sectional area, best-fit diameter (Fig. 5), ellipticity and axonal tortuosity (Fig. 6) have been performed along each axon for all the axons in every 3D reconstructed sample. The measurements of the cross-sectional area and the best-fit diameter are shown in Fig. 5a, b and they appear to follow a lognormal distribution when plotted in terms probability density[44,67]. From the fitted lognormal distributions, the mode and median values of the different zones have been extracted to characterise the underlying probability density distribution.

CC and CR both show a higher concentration of axons characterised by a cross-sectional area of $\sim 0.5\ \mu m^2$, while FO appears to have a larger spread of axons with a modal area of $\sim 1\ \mu m^2$, nearly doubling the CR measurements (Fig. 5a).

The distribution of axonal diameter is similar between CC and CR with peaking values ranging between 1 μm and 2 μm. while FO shows a broader distribution around peaking values of 2 μm. CR and CC share a very similar diameter distribution, with modal values being respectively around 1 μm and 1.07 μm. FO shows the same distributions but with values spread around a higher mode of about 1.4 μm (Fig. 5b). Ellipticity has been calculated as per the formula shown in the inset to Fig. 6a. For $E = 1$ the major axis is double the minor axis of the ellipsis, for $E = 0$ the shape is perfectly circular with the major and minor axes coinciding (Fig. 6a). In the CC over 90% of measured axons shows an ellipticity between 0.6 and 0.9; the values obtained for CR show a similar trend with a slight skewness to large values. FO displays axons that are more elliptic than the values for CC and CR, with numbers in the range of 0.7–0.9 (Fig. 6c).

The tortuosity of the axons has been measured *via* Mimics following the formula also shown in the inset to Fig. 6a. A smaller value of tortuosity corresponds to straighter axon. The measured values give an average tortuosity for the CC of $0.113 \pm 0.109$, for the CR $0.091 \pm 0.07$ and FO of $0.130 \pm 0.088$. Additionally, a comparison of the three areas has been done by plotting the logarithmic values from each group against a theoretical normal distribution (Fig. 6b). All the groups seem to follow a lognormal distribution, but while CC and CR show a similar behaviour, FO differs slightly, in agreement with the general trend depicting some variations when comparing CC and CR against FO.

**Three-dimensional reconstruction and virtual representative volume**. As per microstructural appearance, the 3D reconstruction of the samples analysed shown in Fig. 7, whose data and analysis protocols are provided in full in the Supplementary

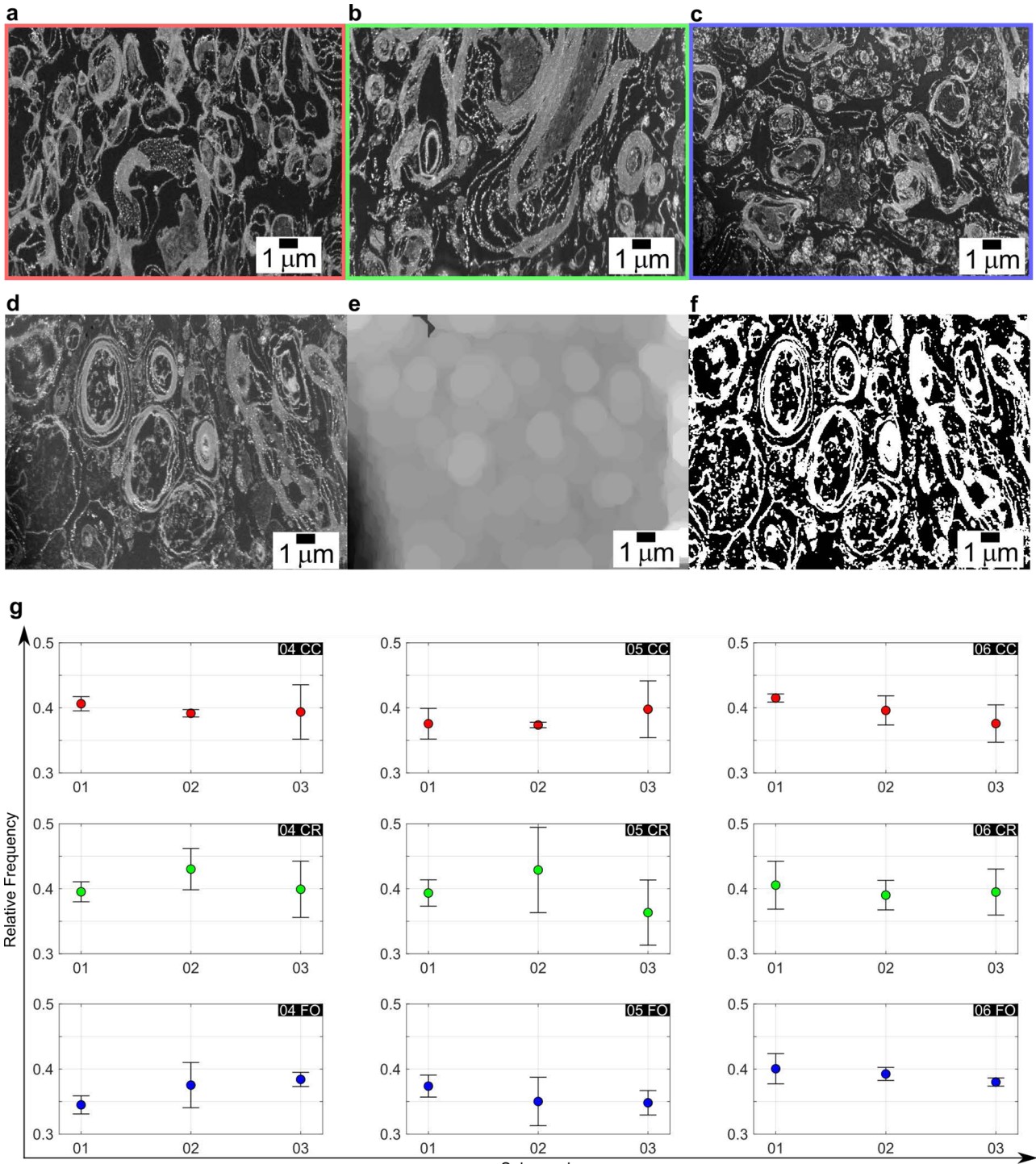

**Fig. 3 FIB-SEM acquisitions, processed slices and results. a–c** The three areas, CC in red (**a**), CR in green (**b**) and FO in blue (**c**), do not appear at first significantly different in the 2D images. **d–f** A series of post processing procedures is applied to reduce the noise in the background before the final binarisation operation. The input image (**d**) is filtered and cleaned of the noise, its own background (**e**) is subtracted to it and final binarisation follows (**f**). By binarising the 2D images a quantification of white pixels representing the myelin is performed. **g** The relative frequency of white pixels was calculated in each of the six 2D SEM images from each sample of each subject. The results show a standard deviation from the average values that differ slightly between samples, but recorded values tend to cluster around a relative frequency value of 0.4. This hints at a constant myelination level that is maintained throughout the areas, with the exception of FO showing a slightly lower average frequency when compared to CC and CR.

Information, show a fibrous arrangement following mostly one main direction, with little to no entanglement of the axonal fibres (except for sample 04 CC). The extra-cellular matrix (ECM) permeates all axons[62], with measured intra-axonal distances between ten and a few hundreds of nanometres.

Following the data measured from the 3D reconstruction, representative in silico 3D models of the microstructure have been created via a custom-made code in Matlab, which is fully described in the Supplementary Information (Supplementary Methods) and made available in an open-source repository. Figure 8 shows the virtually

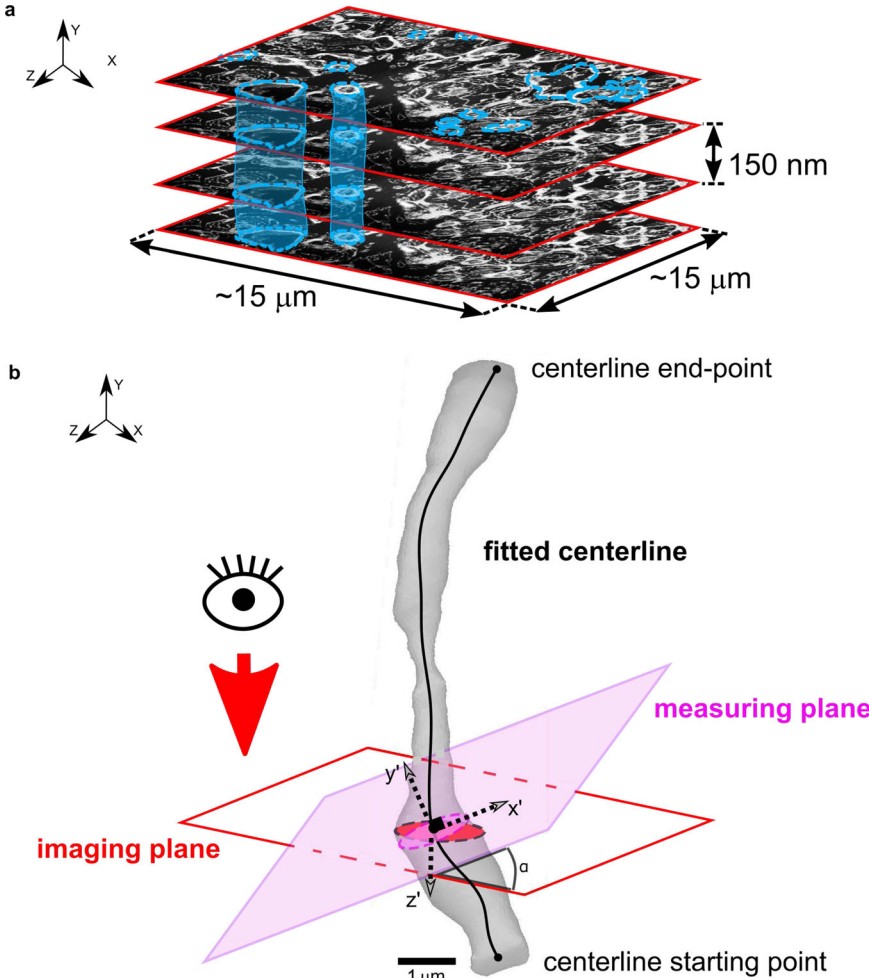

**Fig. 4 Segmentation of the axonal features and measurement strategy. a** Each stack of images is imported into MIMICS[66] where manual segmentation of the axons is performed. The outer axonal perimeter (myelin ring and axonal body) is traced for each axon in each slice imaged every 150 nm. A 2D mask defining the axons (segmented area) is created in MIMICS following the axonal structure in each slice. **b** The software extrapolates a 3D entity which is composed of the interpolated masks and fits a centerline of the axon. The fitting of the centerline ensures the creation of a measuring plane (violet) that is constantly perpendicular to the centerline itself. In fact, the imaging plane is generally not perpendicular to the centerline of the axon due to the impossibility of continuously pre-positioning the axon. Therefore, the variable angle $\alpha$ formed between the two planes can often be the cause of inaccuracies and distortions of 2D measurements made on the imaging plane (red). As measurements are taken on the measuring plane, they are unaffected by the distortion. This method ensures a detailed and precise dataset that represents the whole of the axonal structure with a minimal resolution of 150 nm in the z-direction.

reconstructed RVEs obtained by creating axonal bundles that follow the data distribution of the measured geometrical properties characterising the different cerebral WM tracts: CC is shown in red, CR in green and FO in blue (Figs. 8a–c respectively). Such volumes can be readily imported as STL files into software to be used as the starting point for numerical simulations, e.g. Finite Element or Computational Fluid Dynamics (CFD) models that use Representative Volume Elements (RVEs) to produce microstructurally-based investigations[11,62] or to derive advanced continuum theories[68,69] to study the brain tissue mechanical response to physiological loading. Both these RVEs and the geometric parameters extracted from the image reconstructions can become the input of computational models such as those we have recently presented[59–62].

The use of the virtually reconstructed RVEs for the deterministic modelling of fluid flow or tissue deformation at the micro-scale significantly reduces the computational effort required to determine important quantities (some of which are currently eluding us due to the complexity of reproducing the as-measured tissue cytostructural features in simulations). The underlying probability density distribution obtained for the in

silico reconstructed RVEs shown in Fig. 8d are comparable to those obtained from the quantities measured directly from the 3D tissue reconstruction in Fig. 5. They provide the insight needed to upscale important information to larger scale continuum models. Alternative approaches and other methods can also be used for alternative RVE reconstructions[70].

## Discussion
An engineering approach has guided this study throughout; here the imaging of the features of interest responds to a need of measuring the geometrical properties of these fibrous components (the axons) that heavily influence the mechanical characteristics of WM as a composite material[9,11,71]. FIB-SEM is well suited for an automated and high throughput rate 3D tomographic imaging, especially when the high resolution is not required as the smallest resolvable features are axonal tubular structures with a diameter of 0.2–0.4 μm[33]. Therefore, as this study focuses on the 3D cyto-organization of the axonal fibres for modelling purposes, priority to the output quantity has been

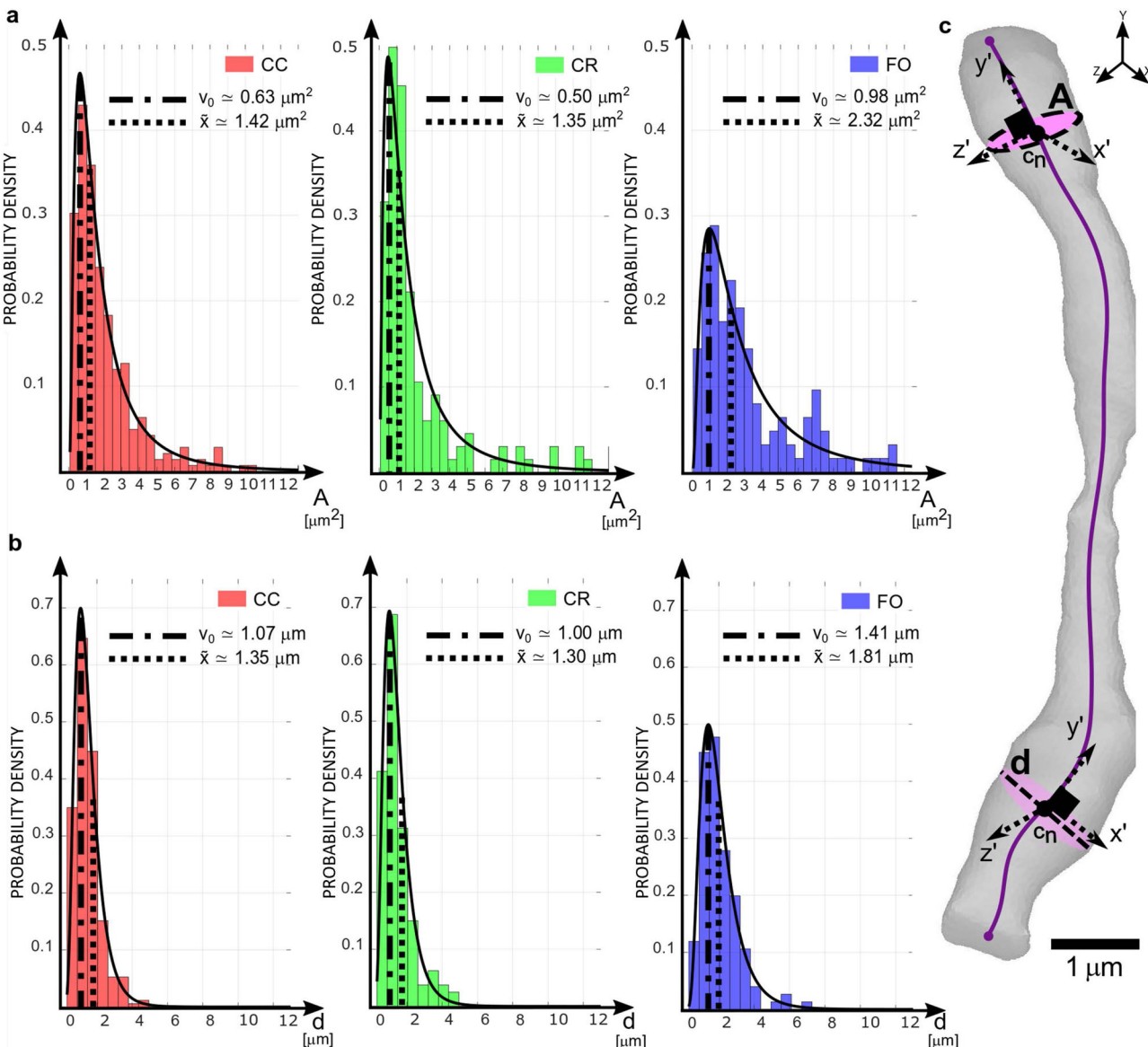

**Fig. 5 Measurements of the 3D axonal structures: area and best-fit diameter. a–c** Each axon is measured every 150 nm throughout its whole length creating an axon-specific database for every axon, from every area, from every subject of the best-fit diameter and the outer axonal ring. First, a fitting of the lognormal distribution over each of the axon-specific database is performed. Then, parameters μ and σ describing the lognormal distribution are extrapolated from each distribution. From μ, is calculated the corresponding arithmetic mean, and from σ the arithmetic standard deviation. These numbers are compared to the arithmetic mean and the arithmetic standard deviation calculated directly from the axon-specific database; the agreement between these two sets of parameters confirms that the lognormal distribution is a good choice for the data representation (Supplementary Information). Then, the average values of area and diameter of each axon are categorized by CC, CR and FO. **a, b** The absence of multiple peaks in the histograms of the cross-sectional areas and the best-fit diameter, suggests the little effect of a possible inter-subject variance. Lognormal distributions are fitted on the data and from their parameters, the values of mode $v_O$ and median $\bar{x}$ values are calculated. **a** Cross-sectional areas in CC presents modal values of 0.63 µm² and median value of 1.42 µm², CR shows modal value of 0.5 µm² and median value of 1.35 µm², FO exhibits a modal value of 0.98 µm² and a median value of 2.32 µm². **b** The best-fit diameters have in CC a modal value of 1.07 µm and a median value of 1.35 µm, in CR a modal value of 1.00 µm and a median value of 1.30 µm, lastly in FO a modal value of 1.41 µm and median value of 1.81 µm are measured. Overall, the distributions of the cross-sectional areas and best-fit diameter, measured as in **c**, show a lognormal behaviour skewed to the lower ends of the scale values for all of the three sampling areas. **a** The majority of the measurements describe axons in CC and CR with a cross-sectional area peaking around the 0.2–1.00 interval while FO shows a wider distribution with axons exhibiting a much wider area between the 0.9–3 interval. **b** Consequently, the best-fit diameters display similar trends for CC and CR behaviour, and different values for FO, which is characterised by larger axons.

given at a cost of a lower resolution of the axonal ultrastructure comprising the extra-cellular matrix. Histological samples used in the present study were preserved using aldehyde-based fixation and dehydration. The cryo-preservation techniques have been shown to produce a limited reduction of extra-cellular space, while the axonal organisation and geometry were mainly intact[72].

Therefore, an aldehyde-based dehydration method provides a good compromise between this need of practical simplicity and the fidelity of the acquired results[44,73]. Another important issue to consider is the minimisation of the impact of artefacts related to post-mortem delay[48,74]. Although this is particularly important for human samples, whose availability and post-mortem handling

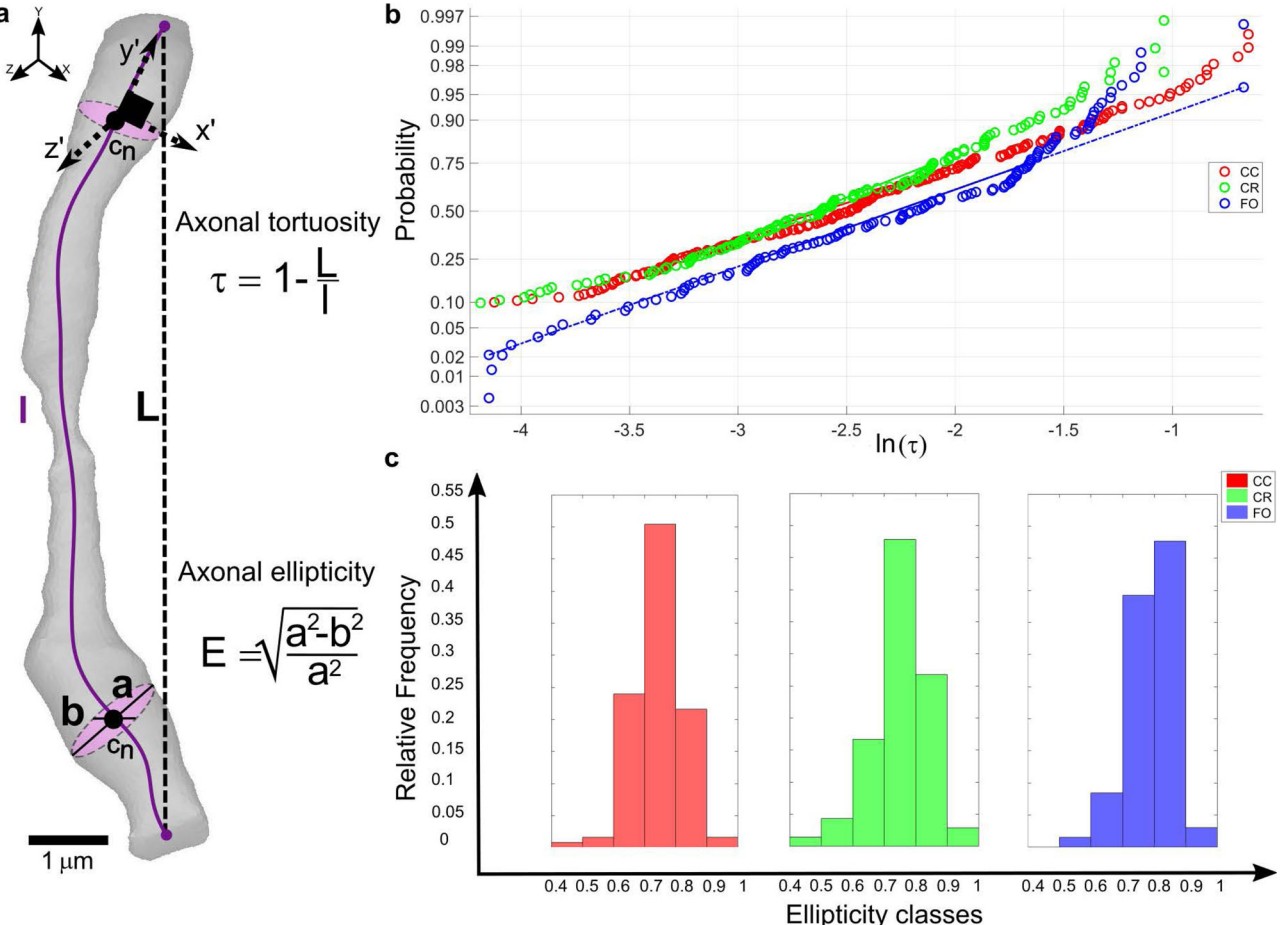

**Fig. 6 Measurements of the 3D axonal structures: Tortuosity and Ellipticity. a** The axonal tortuosity, $\tau$, is measured as one minus the ratio between the length of the centerline and the calculated Euclidean length between its two endpoints. The Ellipticity is calculated as the ratio between the two semi-axes of the fitted ellipsoid. This number quantifies how much the axons appear in a straight fashion. **b** The tortuosity values are calculated per each axon in all the samples and are categorized by area (CC in red, CR in green and FO blue) and their logarithmic value is calculated. This was performed because if $\tau$ is log-normally distributed, then $\ln(\tau)$ has a normal distribution. Therefore, their quantiles are plotted via a Q-Q plot against the quantiles of a theoretical normal distribution (dashed blue line) to see whether CC, CR and FO tortuosity values are log-normally distributed and how they differ from each other. CC, CR distributions show with some deviation from the theoretical normal distribution on the right end side of it. Interestingly, FO seems to reflect its differences with the other two areas also in these measurements with its distribution misaligned from the other two and a much more different behaviour towards the right end side of the graph. **c** It is common to represent the axons as cylindrical entities but the ellipticity values measured here show axons that appear rather elliptic in all of the areas with most of the relative frequency concentrated between the ellipticity values of 0.6 and 0.9, with the FO having a more accentuated elliptical shape confined between the higher values of 0.7 and 0.9.

may significantly affect the preservation of the cytostructures, care must be taken also when excising and preparing the samples to be imaged after animals culling; particular attention has been paid to this as described in the *Methods* sections. This issue further highlights need to link post-mortem and live tissue studies[56], especially given the enormous advances recently made in techniques to study live tissues.

Let us turn our attention more closely to the results obtained in terms of myelin content. Axonal features are bounded by the outer ring of myelin sheaths, so all the data expressed in this research are representative of the "outer diameter" of the axon. Distributions of data follow the findings of previous studies involving ultra-strong gradients for diffusion MRI and different electron microscopy techniques[44,57,67,75]. The authors are aware that white pixels as obtained from the image processing technique proposed here do not represent solely the myelin rings. In fact, for example, some images inevitably contained neuroglia cellular bodies (see yellow dashed profile in Fig. 1a). Cells such as the astrocytes are major cells found in the neuroglia of the CNS[76,77]; therefore, their contribution to the pixel count could not always

be avoided. However, the relative frequency of white pixels still provides a good indicative measure of the myelinisation level of the area. In fact, the value of around 40% of white pixels is in accordance to the relative lipidic content measured in dry weight of white matter samples, which has been reported in an approximate range between 40% and 60%[78]. These numbers are largely determined by the myelin sheaths as dry weight of myelin itself consists of percentages around 70–80% of different lipids i.e. glycosphingolipids, cholesterol and other long-chain fatty acids[79]. Another study, estimated the dry weight content of myelin from WM of adult rats to be around the same value of 40%[80], which further corroborates our findings. This also confirms the chosen size of the ROIs to be representative of the material composition of WM. Additionally, this measured homogeneity of material composition in different, randomly chosen areas across each sample, supports the goodness of the choice of random sampling locations of the areas that have undergone 3D reconstruction.

As previously mentioned, all the samples come from the CNS where the Oligodendrocytes contribute to the lipidic content by forming the myelin rings typical of the WM[77,81,82]. So, the

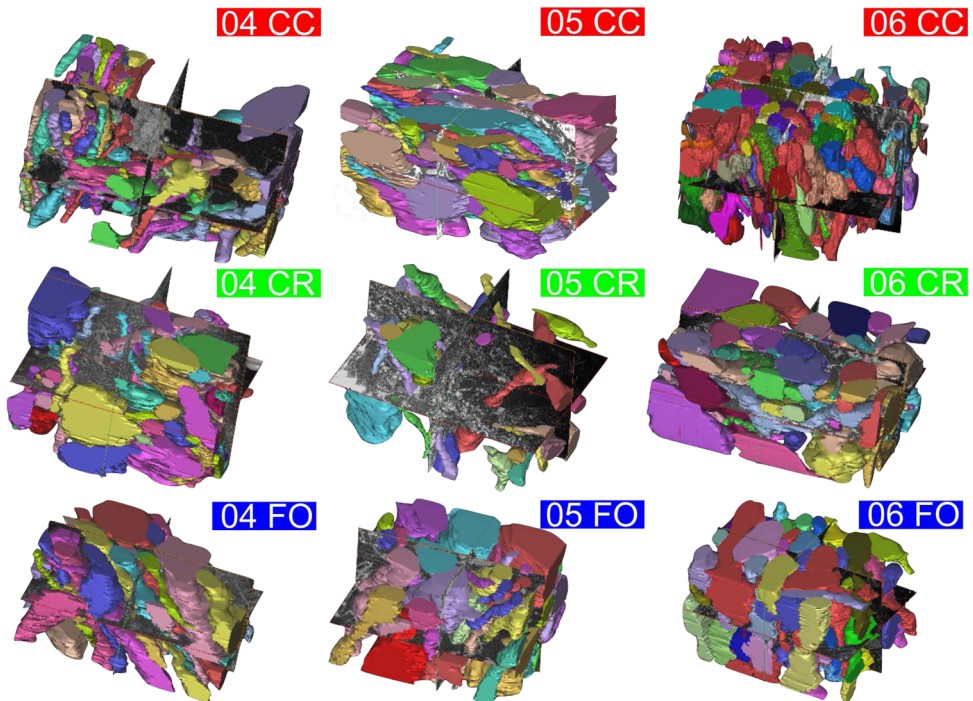

**Fig. 7 Cytoarchitecture of the FIB-SEM 3D reconstructed volumes.** The samples appear to form unidirectional bundles in the majority of the cases, confirming axonal structures ordering for all WM areas considered in this investigation. CC shows an organized cytostructure with sample 04 CC showing the least unidirectional structure of the axons (this is likely due to the sampling being performed at the edge of an axonal bundle). CR presents the lowest unidirectional behaviour in all samples, when compared to the other areas. The fornix exhibits a high unidirectionality in all of the samples (04 FO, 05 FO, and 06 FO) and it also displays the broadest distribution of axons dimensions for the three areas; this could also be due to the more prominently degenerated layers of myelin. The axonal density, calculated as number of axons over the scanned volume, ranges between the areas. CC shows the highest axonal density of circa $2 \times 10^7 \pm 5 \times 10^6$ axons/mm$^3$, the CR of $1.5 \times 10^7 \pm 5 \times 10^6$ axons/mm$^3$ and the FO of $1.1 \times 10^7 \pm 3 \times 10^6$ axons/mm$^3$. In more engineering terms, the Volume Fraction has been calculated as the fraction between the total sum of the volume of the axons and the total volume scanned. FO shows the highest volume fraction of circa $0.51 \pm 0.05$, the CC values around $0.45 \pm 0.12$ and the CR of $0.37 \pm 0.21$.

consistency found among all samples could be explained by the uniformity of the cells involved in the myelination of the axons. Additionally, all the sampled fibre tracts belong to the CNS, here the axons are shorter than the ones in the Peripheral Nervous System (PNS) so a uniform level of myelination would be expected. Oligodendrocytes wrap their membrane processes around the axons. Therefore, EM images show myelin sheaths as periodically layered areas of electron dense and light dense regions that represent the alternation of intra-period and dense lines (see red dashed insert in Fig. 2e). This feature is where the oligodendrocitic membrane comes into apposition to form the isolating layer[83–85]. Therefore, the recognition of this feature aided in the correct individuation of myelin rings also in the eventuality of split, degenerated layers.

Values of the CC are within the range of the existing literature, with >50% of the measurements in terms of axon diameter falling in the range from 0.5 μm to 1.5 μm[44,86–89]. Interestingly, the values measured in ovine brain are closer to the ones measured in chimpanzees and humans than the measurements taken from macaques. In fact, although the range of values and the lognormal distribution of data is maintained, the spread of the ovine values is more like the former two species than the latter, which exhibits a higher concentration of smaller axons in the CC. Therefore, this study, which has been performed analysing a number of representative cross-sections of different fibre tracts in analogy to studies reported on other species, also shows that ovine brain can be successfully used as an animal model to investigate biomedical problems strongly linked to the human cerebral cytostructure[90]. Additionally, the range of axons detected in each area, although differing slightly between each other, are all within the range of

quantities measured in other studies[86,91,92] with the CC showing an axonal density of circa $2 \times 10^7 \pm 5 \times 10^6$ axons/mm$^3$, the CR of $1.5 \times 10^7 \pm 5 \times 10^6$ axons/mm$^3$ and the FO of $1.1 \times 10^7 \pm 3 \times 10^6$ axons/mm$^3$.

CR shows a trend very similar to the CC but with smaller axons. To the authors' knowledge, FO values have never been systematically measured before and therefore comparison with literature is hindered by a lack of data. However, the higher values measured are also due to the numerous irregularities and degeneration of the myelin layers that have been found in the FO samples. These degenerative patterns have been widely found in several studies across different species and although their consequences on the brain functionality have not yet been fully understood, it has been related to the normal aging processes occurring in all living organisms[93,94].

A bigger scatter of data is noticeable when comparing the FO to the CC and CR. The variation of the diameter and the cross-sectional area measured within the same axonal structures is due to the different subcellular structures within each axon. The most prominent is the mitochondria, which appears in the form of bulges along the axonal tube[57] (Fig. 5a). Ellipticity findings in all of the three areas, unequivocally show that axons should be thought as ellipsoidal tubular structures[57] with ellipticity values comprised between 0.7 and 0.9, rather than circular as commonly assumed by the biomechanics community[9–11,95,96].

This study also gives the first quantitative measure of the level of tortuosity of the fibrous component of the WM. Interestingly, also this geometrical quantity follows a lognormal distribution. Low values of tortuosity from all of the three fibre tracts show that axons follow a nearly straight path in the CNS at this scale of

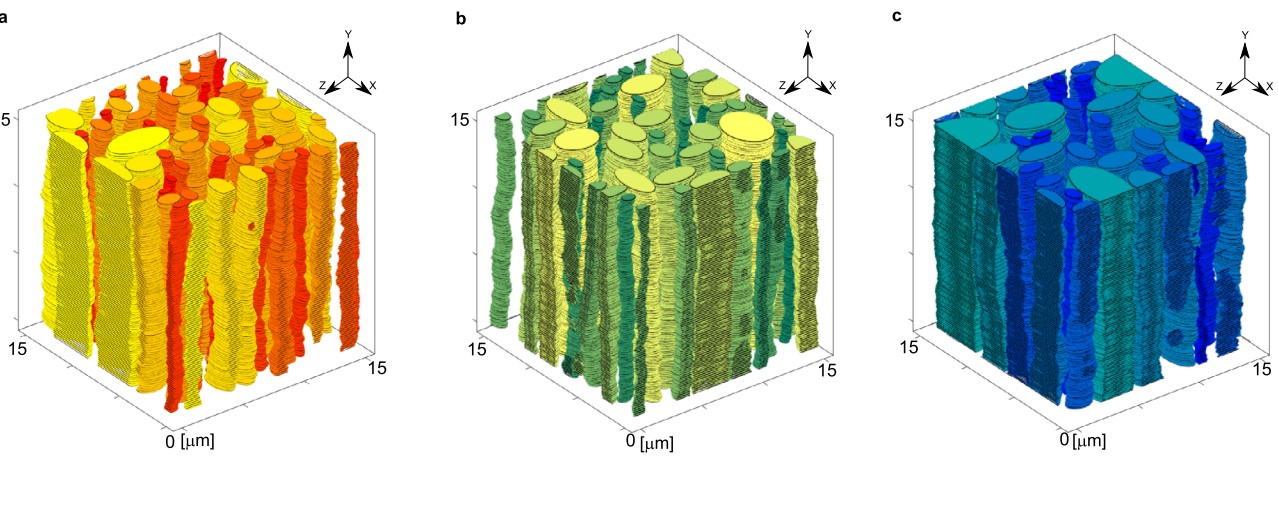

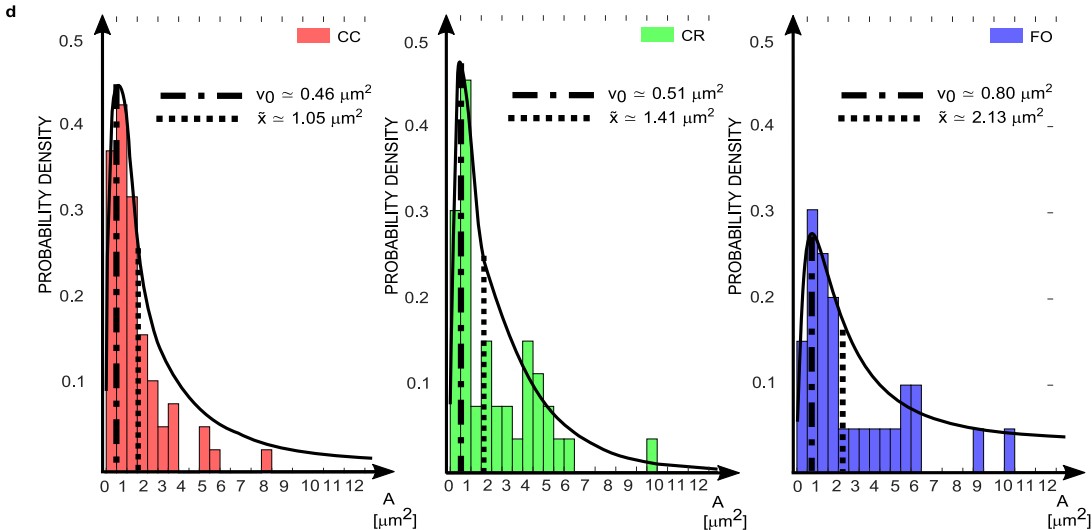

**Fig. 8 3D Representative Volume Elements of WM tracts generated in silico. a–c** Three volumes of $15 \times 15 \times 15\,\mu m^3$ were virtually reconstructed for CC (in red), CR (in green) and FO (in blue). These "in vitro" WM volumes are representative of the three different zones in terms of cross-sectional area, cross-sectional diameter, tortuosity, axonal density and volume fraction. The input data (a dataset per each WM tract) for the code consists in: the parameters $(\mu,\sigma)$ of the data distribution of the area, the minimum and maximum ellipticity value, the average axonal density, the average volume fraction and its standard deviation, the interlayer variability of the measured area, the average interlayer tortuosity and its standard deviation and the interlayer difference in semi-axis length. This overall dataset has been extrapolated from the experimental dataset as all the aforementioned quantities are saved per each slice, per each axon, per each volume in each subject. **d** The underlying probability density distribution obtained for the in silico reconstructed RVEs. The reconstructing algorithm is given some freedom in the creation of the volumes in order to accommodate the axons to follow the distributions of the experimental measurements. The volumes are created in order to be periodic in the $x$–$y$ direction and therefore are ready to be imported as STL files into commercial Finite Element Analysis software (e.g. ABAQUS[110]). The commented source codes are provided in the Supplementary Information, where more details can be found.

investigation. Here, a difference between the FO and the CC and CR seem to suggest that while commissural and projection fibres have a more similar architecture, the long projection fibre of FO seem to have a higher level of tortuosity, probably due to its particular anatomical configuration. The macroscopic shape of the FO, with its more accentuated bending and curving, might be the reason of the measured difference. It is important to highlight that previous studies always took into considerations WM samples from cranial nerves, brain stems or from spinal cord where a much wavier layout of the axonal tracts is visible[9–11,71].

Finally, it must be recognised here that manual sampling from visual inspection of the areas of interest does not provide a precise determination of the sample excision location. Therefore, the additional use of MRI images of the sampled brains as reference images will lead to important improvements when determining the sampling position relative to the axonal tracts of the cerebrum[97]. Cryo-imaging techniques have also been shown to better preserve the ultrastructure of the biological matter when compared to the alcohol-driven dehydration methodology followed in this study[84], which might have also resulted in limited shrinkage of the sampled tissue[98,99], here minimised through the implementation of a well-controlled and gradual dehydration procedure (see *Methods*). Despite this potential limitation, the room temperature fixation technique herein adopted, provides a robust, easy and rapid solution for high-throughput imaging of multiple specimens. The comparisons with other studies and the

agreement with values obtained from different imaging techniques[44,67,75,91,100] show that any adverse effect on the structure does not significantly affect the overall reconstruction of the tissue microstructure and that the geometrical properties of the axons here presented are reliable.

## Conclusions

This study presents a first systematic attempt at characterizing different pathways of brain WM and it shows the feasibility of the proposed methodology for the creation of an axonal database for each of the tracts present in the WM. At the microscale, all of the investigated WM pathways, the CC, the CR and the FO, appear as unidirectional fibrous composites with axons being relatively straight, elliptical tubular structures and embedded in an isotropic matrix composed of ECM and accessory cells. Such a detailed knowledge of the brain microstructural features and their properties can provide key insights and unlock a deeper understanding of the different phenomena and properties, such us local diffusivity, permeability and stiffness, that govern the macroscopic behaviour of cerebral tissue and its mechanical response to physiological and external stimuli as well as axonal conductivity in different areas of the brain.

## Methods

**Samples preparation**. Three healthy, female *ovis aries* (1 year old, 70 Kg weight) have been used for this study. All animals have been treated under the European Communities Council Directive (2010/63/EU), adhering to the laws and regulations on animal welfare enclosed in D.L.G.S. 26/2014 and it has been approved by the Italian Health Department with authorization n° 635/2017. After culling *via* intravenous potassium chloride overdose, following the authorization n° 635/2017, the cerebrum has been immediately removed. CC, FO and CR have been sampled using biopsy punchers of 1 mm diameter from the middle coronal section of each cerebrum (see schematic in Fig. 2a, b). Given the theory of cerebral laterality[44,101,102], a conservative choice of sampling only from the left hemisphere has been taken to avoid any possible difference between left and right hemisphere. Additionally, to ensure a higher precision and a consistent repeatability in the sampling area, a custom-made brain slicer has been developed. A cast of excised sheep brain underwent 3D scanning and its resulting CAD model has then been used to design a stainless steel sectioning matrix with equidistant blade guides, obtained *via* laser-cutting, per each mm in the axial and sagittal directions.

A simple and robust preparation protocol has been followed in order to obtain a high-rate imaging with a satisfactory contrast for the measurement of the outer axon structure only. Therefore, the staining and embedding method from Mikula and Denk[103] has been followed with few modifications[104]. Within 20 minutes from culling, each biopsy sample has been fixed in 2% formaldehyde in 0.1 M sodium phosphate buffer (pH 7.4) for 3 h at room temperature. After fixation, the samples have been washed three times for 10 minutes in the buffer and left at circa 4 °C overnight to remove any remaining fixative within the tissue.

The samples have been stained in 0.5% (w/v) osmium tetroxide ($OsO_4$) in 0.1 M Phosphate Buffer Solution (PBS) for 1 h to make visible the myelin layers as it binds to the lipidic layers[105]. Three extra washes of the samples, each one of 10 minutes in the buffer, ensure the removal of excessive stain. Samples have been gradually dehydrated *via* immersion in ethanol-water solutions, namely at 25%, 50%, 70% and 95% for 15 minutes each, in order to minimise trauma to the tissue and reduce the impact of its shrinkage. Finally, samples have been immersed two times for 15 minutes each, in fresh 100% ethanol.

Embedding in LRWhite resin has been gradually performed in three steps. First, samples had been left in a 1:1 solution of ethanol and resin for 2 h. Then, samples have been immersed in another 1:3 solution of ethanol and resin for 2 h. Finally, samples have been left in pure LRWhite resin overnight. Anaerobic thermal curing in oven (60 °C for 24 h) produces the final samples ready for sectioning.

From each embedded sample, three subsamples (see schematic in Fig. 2c) have been excised *via* microtoming (PTXL PowerTomes, RMC Boekler) using a glass knife. Then, each sample has been mounted on a stub and gold coated for the subsequent FIB-SEM imaging.

**FIB-SEM Imaging**. Imaging has been performed *via* a Zeiss Auriga Cross Beam featuring a Schottky field emission gun and a Gemini electron column. The SEM column is coupled with a Ga+ ion FIB.

FIB-SEM series have been taken with FIB milling conditions set to 30 kV and with a beam current of 4 nA corresponding to 150 nm thick slices, on samples tilted at 54°, as per FIB-SEM manufacturer instructions. The slice thickness was chosen to optimise the acquisition time while still being able to study and analyse relatively large volumes of several samples and develop a comprehensive database. Preliminary analyses, not reported here, were performed to reconstruct smaller samples using slice thicknesses ranging from 10 nm to 200 nm and comparing the 3D reconstruction obtained from such samples. Minimal differences were recorded in terms of the key geometrical parameters reported above for slice thicknesses up to 175 nm; however, a progressive loss of the ability to resolve features in the directional orthogonal to the slices was identified with increased slice sizes. While this may be a limitation in terms of the use of the images obtained here in terms of overall resolution, especially in the context of their use with semi-automated or automated segmentation procedures, for which thinner slices are recommended, this enabled to optimise acquisition time, therefore also preventing potential issues associated with loss of focus and alignment for long-term image acquisition[40].

SEM imaging has been performed via a beam energy at 1.5 kV to avoid damage of the tissue sample and polymerisation of the embedding resin. Selective backscattered electron detector has been used to image the contrast given by the heavy metal $OsO_4$ stain, now bound to the lipidic layer of the myelin[106]. 16-bit Black and White have been acquired *via* the detector and stored with a resolution of $764 \times 1024$ pixel-sized images characterised by a resolution of 0.020 μm/pixel. Tilting correction and line averaging noise reduction was applied by the in-built acquisition software before final image acquisition and storing. On average, the total acquisition time resulted in ~24 h per each 3D (15 μm$^3$) volume. All images are included in the Supplementary Information and are made available in an open-source repository.

**3D reconstruction, measurements and in silico RVEs generation**. After FIB-SEM images have been randomly acquired from each sample, Each stack of FIB-SEM images representing a 3D sample has been post-processed *via* Mimics software (Materialise, 2019). First, a manual segmentation of each axon present in the region of interest had been carried out. Then, using the built-in 3D reconstruction algorithm of MIMICS, 3D models have been created by interpolating the different layers (e.g. Fig. 7). Therefore, from each subject, a full 3D reconstruction of CC, FO and CR has been created following the procedure detailed in the Supplementary Information, with codes and data made available in an open-source repository.

Each axon was then analysed and measurement were taken as summarised in the Supplementary Information (See Supplementary Methods—3D reconstruction and measurements: post-processing), where cross-sectional area values obtained from the reconstructed axons and their analysis are also reported in Supplementary Fig. S1 and S2. Note that the manual sample segmentation and the 3D reconstruction can also be used to test and validate the use of automated and semi-automated methods to obtain 3D reconstructions using the set of images provided by the authors to test the accuracy of the proposed methods.

The in silico generation of the 3D RVEs, whose typical results are shown in Fig. 8, has been performed using the algorithm provided by the authors as part of the Supplementary Information and schematically shown in Supplementary Fig. S3.

**Reporting summary**. Further information on research design is available in the Nature Research Reporting Summary linked to this article.

## Data availability

All images and data have been made available via a Zenodo repository: Bernardini, A., Trovatelli, M. & Dini, D. EDEN2020 Ovine Brain Dataset for Imaging and Reconstruction of the Cytoarchitecture of Axonal Fibres (Version 1.0) [Dataset]. Zenodo (2021), https://doi.org/10.5281/zenodo.4772440. Further explanations and information about accessibility and interpretation of the data are also provided in the Supplementary Information file. The source data behind the graphs in the paper have also been provided as part of the Supplementary Information (Supplementary Data 1).

## Code availability

All codes have been made available via a Zenodo repository: Bernardini, A., Trovatelli, M. & Dini, D. EDEN2020 Ovine Brain Dataset for Imaging and Reconstruction of the Cytoarchitecture of Axonal Fibres (Version 1.0) [Dataset]. Zenodo (2021), https://doi.org/10.5281/zenodo.4772440. Further explanations and information about accessibility and interpretation of the codes are also provided in the Supplementary Information file.

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

## Acknowledgements
This project has received funding from the European Union Horizon 2020 research and innovation programme under Grant Agreement No. 688279. Daniele Dini acknowledges the support received under the EPSRC Established Career Fellowship Grant EP/N025954/1 and the Shell/Royal Academy of Engineering Research Chair.

## Author contributions
A.B., D.D. conception or design of the work; A.B., M.T., M.M.K., A.P. acquisition, analysis, and interpretation of data; A.B., M.P., D.D. creation of new software used in the work; D.D.Z., S.B., A.P., F.R.y.B., D.D. supervision, research resources and funding; all authors drafted the work or substantively revised it.

## Competing interests
The authors declare no competing interests.

**Additional information**

