## [Peer Review File · Communications Biology]

Reviewers' comments:

Reviewer #1 (Remarks to the Author):

This manuscript describes detailed characterisation of axonal cytoarchitecture in the ovine brain, with consideration of commissural, association and projection fibres. The authors approach the data from 2D and 3D perspectives, first examining myelination via FIB-SEM and comparing between corpus callosum, corona radiata and fornix, and then performing more detailed 3D analysis of axonal tortuosity and ellipticity in each tract, and finally modelling the ellipticity, axonal density and volume fraction. This work is novel and the authors have performed a comprehensive analysis.

I have some minor comments:

- The authors should specify why ovine samples were used and why 3 animals was deemed sufficient.
- It would help the reader to understand in the introduction (or even earlier in the methods) the practical advantage of slicing the tissue in the way they have, to enable white matter punches from 3 tracts on the same slice.
- More information is needed on the 3D reconstruction, particularly on slice spacing and alignment.
- Abstract: "first systematic attempt to characterise..." this might need rewording. First using SEM perhaps, and/or first to characterise simultaneously at this level.
- Abbreviations for tracts have been mentioned in the introduction - no need to re-define them later.

Reviewer #2 (Remarks to the Author):

Title: Imaging and reconstruction of the cytoarchitecture of axonal fibres: enabling biomedical engineering studies involving brain microstructure

This paper provides a new approach to the characterization of tissue for engineering purposes, and it gives a useful insight into the 3D fibrous structure and organization of the WM. It is interesting that FIB-SEM was successfully acquired – I was not sure this could be done. I had always heard it would be too destructive to fib tissue. The paper used novel testing and experimental procedures and statistical reporting methods were used. The paper makes a valuable contribution to the literature. In the intro you make a case for drug delivery, but this provides useful information for brain injury studies too. The paper was clearly written, and I enjoyed reading it. I think the paper can be accepted as it is, but some minor modifications might help strengthen it for the reader.

Below are my suggestions and comments:

- For Figure 1, you might want to also label the images as macro, meso and microscopic levels.
- Fig. 2 are most nerves about 1 um in diameter? How representative is the cross-section shown?
- Quite a bit of time is spent analyzing and discussing the relative frequencies but it is not clear why that metric is so important. Further motivation would be helpful here.
- Fig 5 is very powerful.
- Biggest issue is how the post-mortem results differ from live tissue. I completely support the approach, but I have seen PM tissue that is fixed, and it seems more "orthotropic" or aligned. I have often wondered If it is the same in living tissue.
- Just to be clear, you refer to fibers. Are fibers the same fibers we see in DTI. What is the difference? Also, how many axons are in a fiber? Do you know that?
- Why do the in silico RVEs (fig 8) look so different from the fib-sem 3d reconstructed volumes (fig 7)?
- What sort of damage does the FIB do to the tissue? Is it altering your geometries some how. It appears not to.
- How large a sample can you do? Would you fib entire brain?
- Is there any conceivable way to put one of these RVEs under mechanical loading?

- Are there any noticeable intra-RVE connections or networks that can be mapped out?

Thank you for allowing me to review this paper. It is really an impressive work.

Reviewer #3 (Remarks to the Author):

In this work, the authors study the brain microstructure of the white matter in ovine samples. For that, they used 3D-electron microscopy to reconstruct myelinated fibers in selected areas of the brain: the corpus callosum, coronal radiata and fornix. Using the segmented axons, the authors estimate parameters of interest associated with the axonal morphology.

1. The authors present in this study "the first systematic attempt..." to characterize the microstructure of white matter. However, many studies have already described the architecture of the white matter using a variety of modalities from traditional 2D light and electron microscopy to advanced 3D light and electron microscopy or x-ray tomography between other methodologies, e.g. polarized light microscopy or coherence tomography. The authors must revise previous studies and acknowledge the evolution of the methodology from traditional to advanced in the context of white matter. In the current version of the manuscript, it is not clear what this study add to the study of white matter.

2. A big limitation in this study is that the authors introduce a manual approach for axonal segmentation, while several studies have been already developed automated and semi-automated segmentation methods for the white matter for higher and lower image resolution as acquire in this work. Few examples of latest published studies on 3D electron microscopy or x-ray tomography offer interesting pipelines for segmentation and quantification of white matter samples:

Along-axon diameter variation and axonal orientation dispersion revealed with 3D electron microscopy: implications for quantifying brain white matter microstructure with histology and diffusion MRI. Lee HH, Yaros K, Veraart J, Pathan JL, Liang FX, Kim SG, Novikov DS, Fieremans E. *Brain Struct Funct.* 2019 May;224(4):1469-1488. doi: 10.1007/s00429-019-01844-6. Epub 2019 Feb 21.

Automated 3D Axonal Morphometry of White Matter. Abdollahzadeh A, Belevich I, Jokitalo E, Tohka J, Sierra A. *Sci Rep.* 2019 Apr 15;9(1):6084. doi: 10.1038/s41598-019-42648-2.

Axon morphology is modulated by the local environment and impacts the noninvasive investigation of its structure-function relationship. Andersson M, Kjer HM, Rafael-Patino J, Pacureanu A, Pakkenberg B, Thiran JP, Ptito M, Bech M, Bjorholm Dahl A, Andersen Dahl V, Dyrby TB. *Proc Natl Acad Sci U S A.* 2020 Dec 29;117(52):33649-33659. doi: 10.1073/pnas.2012533117. Epub 2020 Dec 21.

DeepACSON automated segmentation of white matter in 3D electron microscopy. Abdollahzadeh A, Belevich I, Jokitalo E, Sierra A, Tohka J. *Commun Biol.* 2021 Feb 10;4(1):179. doi: 10.1038/s42003-021-01699-w.

A semi-automated approach to dense segmentation of 3D white matter electron microscopy. Michiel Kleinnijenhuis, Errin Johnson, Jeroen Mollink, Saad Jbabdi, Karla L. Miller. doi: <https://doi.org/10.1101/2020.03.19.979393>

The authors must review the literature regarding segmentation and quantification of brain samples using 3D methodologies, and demonstrate if their method is more efficient or consider using a more advanced method. This also affect the quantification: the present work is quite limited in parameters calculated from the axons, while other methods can provide more information of these structures.

3. The authors claim on pg 5 that the acquisition protocol was optimized to enable large volumes. 15 μm^3 is rather small volume in terms of brain microstructural characterization as shown in the papers above.

4. Are the authors confident that all "white pixels" come from myelin? Are the removed "white pixels" not crucial to estimation of myelin in these data? As seen in Figure 3, there is delamination of the

myelin sheaths which can affect the estimation of the diameter of the axons + myelin (what the authors call outer diameter of the axon), estimation of g-ratio (not calculated in this study) or in pathological conditions (such as demyelination). If this is a healthy animal, can the authors explain the delamination of the myelin sheaths?

5. The thickness of the slices during imaging was 150 nm. This is a thick slice in terms of microstructural characterization. This should be addressed as a limitation in the discussion.

6. Shrinkage with chemical fixation is not negligible as the authors claim on page 16. This has been extensively shown, e.g.:

Ultrastructural comparison of dendritic spine morphology preserved with cryo and chemical fixation. Tamada H, Blanc J, Korogod N, Petersen CC, Knott GW. *Elife*. 2020 Dec 4;9:e56384. doi: 10.7554/eLife.56384.

7. Because of the shrinkage, the extracellular space is basically gone in chemically fixed samples. Are the authors able to see extracellular space? if so, where? could the authors point at the space in the figures?

Related to that, if the shrinkage minimizes the extracellular space, the reconstructed axons could have very little used to model, for example, drug delivery, as the authors claim in the introduction. This issues must be addressed in the discussion section.

8. The authors must explain in more detail the methodology.

9. The discussion is long, and many points are trivial, e.g. explanation about basic neuroscience.

Response to Reviewers' Comments

Communications Biology, Manuscript COMMSBIO-21-1735-T

“Imaging and reconstruction of the cytoarchitecture of axonal fibres: enabling biomedical engineering studies involving brain microstructure”

Andrea Bernardini, Marco Trovatielli, Michał M. Kłosowski, Matteo Pederzani, Davide Danilo Zani, Stefano Brizzola, Alexandra Porter, Ferdinando Rodriguez y Baena, Daniele Dini

We would like to thank the reviewers for their extremely valuable comments and suggestions. We have found the recommendations very insightful; they have certainly allowed us to significantly improve the readability and the quality of our article. In the following, we have addressed all reviewers' comments and suggestions. The main modifications are indicated in yellow in the highlighted version of the revised manuscript. The revision also led to the addition of several new references (21), now discussed as part of our response.

We apologise for the delay in returning the revised manuscript, which has been due to unforeseen circumstances linked directly and indirectly to the COVID pandemic and event which affected some of the authors of the manuscript.

We look forward to hearing from you about the final stage of the review process.

Reviewer #1 (Remarks to the Author):

This manuscript describes detailed characterisation of axonal cytoarchitecture in the ovine brain, with consideration of commissural, association and projection fibres. The authors approach the data from 2D and 3D perspectives, first examining myelination via FIB-SEM and comparing between corpus callosum, corona radiata and fornix, and then performing more detailed 3D analysis of axonal tortuosity and ellipticity in each tract, and finally modelling the ellipticity, axonal density and volume fraction. This work is novel and the authors have performed a comprehensive analysis.

Authors' response: We would like to thank the reviewer for the positive assessment of our work and for appreciating the novelty of the proposed approach and the comprehensive analysis performed in our work.

I have some minor comments:

Comment

- The authors should specify why ovine samples were used and why 3 animals was deemed sufficient.

Authors' response: Many thanks for this comment, which has enabled us to clarify some of the main assumption of our study. Ovine samples were chosen as part of the EDEN2020 (<https://www.eden2020.eu/>) study as part of a broader investigation aimed at providing key information and technological advances to improve convection-enhanced drug delivery (CED) procedures. The starting point of the study was the key reference study on humans and macaques (Ref. [48] of the revised manuscript), which also studied three brains. This was deemed to be a good compromise between the (already large) time needed to acquire the images across different regions of the individual animals and the need to investigate intra-subject variations. Our investigation shows that there is minimal variation between the samples obtained from different brains, which corroborates the validity of our initial hypothesis. This has now been discussed in the revised version of the manuscript.

Comment

- It would help the reader to understand in the introduction (or even earlier in the methods) the practical advantage of slicing the tissue in the way they have, to enable white matter punches from 3 tracts on the same slice.

Authors' response: We have now revised both Introduction and Methods to better explain the advantage of slicing the tissue as described in the protocol chosen for this study.

Comment

- More information is needed on the 3D reconstruction, particularly on slice spacing and alignment.

Authors' response: Many thanks for this very useful suggestion. Although slice spacing was already reported in the original paper, we have now extended the Methods section to provide more information both in terms of the 3D reconstruction and the manual segmentation of the images.

Comment

- Abstract: "first systematic attempt to characterise..." this might need rewording. First using SEM perhaps, and/or first to characterise simultaneously at this level.

Authors' response: We have now revised the Abstract according to the suggestions received by this and other reviewers.

Comment

- Abbreviations for tracts have been mentioned in the introduction - no need to re-define them later.

Authors' response: Agreed. Based on this comment, the paper has been revised to avoid repetition when defining the abbreviations after they are reported in the Introduction.

Reviewer #2 (Remarks to the Author):

Title: Imaging and reconstruction of the cytoarchitecture of axonal fibres: enabling biomedical engineering studies involving brain microstructure

This paper provides a new approach to the characterization of tissue for engineering purposes, and it gives a useful insight into the 3D fibrous structure and organization of the WM. It is interesting that FIB-SEM was successfully acquired – I was not sure this could be done. I had always heard it would be too destructive to fib tissue. The paper used novel testing and experimental procedures and statistical reporting methods were used. The paper makes a valuable contribution to the literature. In the intro you make a case for drug delivery, but this provides useful information for brain injury studies too. The paper was clearly written, and I enjoyed reading it. I think the paper can be accepted as it is, but some minor modifications might help strengthen it for the reader.

Authors' response: We would like to thank the reviewer for the very positive feedback provided and for the insightful comments, which have enabled us to significantly improve the quality of the presentation of our work and the flow of the paper.

Below are my suggestions and comments:

Comment

- For Figure 1, you might want to also label the images as macro, meso and microscopic levels.

Authors' response: Figure 1 has been modified accordingly.

Comment

- Fig. 2 are most nerves about 1 um in diameter? How representative is the cross-section shown?

Authors' response: The analysis performed on all samples (and indeed the characterisation performed by other researchers and widely reported in the literature) shows that there exists a distribution of axon's diameters with median values around 1 micron – see Figure 5. This is the results of scans from multiple samples from the same brain regions excised from different brains. As discussed further in the revised version of the manuscript, the cross-section shown is indeed representative of the WM investigated and the size of the samples and SEM images are in-line with those reported and studied in the relevant literature. It is also worth noting at this stage that the values measured for ovine brains here are indeed very similar to those obtained for chimpanzees and human samples, as reported in the Discussion section.

Comment

- Quite a bit of time is spent analyzing and discussing the relative frequencies but it is not clear why that metric is so important. Further motivation would be helpful here.

Authors' response: Thanks for the comment. Here by relative frequency we intended the ratio between number of white pixels (representing lipid content and myelinated areas) and the total number of pixels. This provides a good indication of the myelin content, which in turn can be compared with other existing measurements for examples performed in Refs. [48, 89-92] and discussed later in the paper (Discussion section). However, we agree the terminology used could cause confusion, so we have reworded this and added a note to clarify the use of such relative frequency in this context.

Comment

- Fig 5 is very powerful.

Authors' response: Many thanks!!!

Comment

- Biggest issue is how the post-mortem results differ from live tissue. I completely support the approach, but I have seen PM tissue that is fixed, and it seems more “orthotropic” or aligned. I have often wondered If it is the same in living tissue.

Authors' response: We totally agree with the reviewer, and we understand analyses of the tissue post-mortem will give differences. This is the reason why, in order to minimize the impact of artefacts related to post-mortem delay, sample preparation must be performed both, carefully and efficiently. This issue becomes less problematic when animals are not affected by specific pathologies (as is the case here) are studied. This also prompts the need to link post-mortem and live tissue studies, especially given the enormous advances recently made in techniques to study live tissues. This is now briefly discussed in the revised version of the manuscript.

However, it should also be noted here that the fixation itself is also very impactful, as discussed in our response to reviewer #3. Both these issues are discussed in more details and relevant references are provided in the revised version of the manuscript. It should also be born in mind that post-mortem images are indeed very much needed to study the detailed effect of the microstructure on the tissue’s characteristic response to e.g. deformation induced by differential pressure emerging from pathologies, drug diffusion and neurosurgical interventions.

Comment

- Just to be clear, you refer to fibers. Are fibers the same fibers we see in DTI. What is the difference? Also, how many axons are in a fiber? Do you know that?

Authors' response: Many thanks for this comment. Here by fibres we intend the fibres seen in DTI, i.e. the fibre bundles. Axons are the individual components of fibre bundles. We noted that in one instance we had misleadingly used “fibre” to identify one axonal structure. This has now been corrected to avoid misunderstanding in the use of our terminology.

Comment

- Why do the in silico RVEs (fig 8) look so different from the fib-sem 3d reconstructed volumes (fig 7)?

Authors' response: The differences in the *in-silico* reconstruction of the RVEs in Figure 8 with those reconstructed from the FIB-SEM samples in Figure 7 are due to the fact that in our *in-silico* reconstructions we intended to generate 3D samples which can be readily used for computation of e.g. permeability or tissue mechanical response. This can only be achieved by simplifying the axonal arrangement while still capturing the key features and parameters’ distributions computed from the FIB-SEM samples. We believe that this in itself will significantly reduce the computational effort required to determine important quantities (some of which are currently eluding us due to the complexity of the as-measured tissue cytostructural features) and provide the insight needed to upscale important information to larger scale continuum models. Alternative approaches and other methods conventionally used to reconstruct 3D fibrous microstructures can also be used to obtain alternative RVEs representation – this has now been discussed in the revised version of the manuscript, where we also provide suggestions for further improvements and recommendations for successful use and implementation of the proposed method.

Comment

- What sort of damage does the FIB do to the tissue? Is it altering your geometries some how. It appears not to.

Authors' response: Damage to tissues from the ion beam produces curtains and waves of non-uniform material removal that limit data quality. Here we do not see this damage as the ion beam current here (4nA) is below the threshold that causes tissue damage, as also explained in Ref. [111], now added to the discussion of this point in the Methods section.

Comment

- How large a sample can you do? Would you fib entire brain?

Authors' response: The volume that can be imaged will be limited by the on the time taken to FIB-SEM a whole brain which would depend on the voxel size needed to image features of interest. For example in recent work it took 2 weeks to FIB through a volume of 30x30x60 μm^3 of nerve tissue using 10x10x10 nm^3 voxels. Better resolution requires a longer image acquisition time; conversely, a reduced volume size will be accessible in a fixed time. This concept is very nicely illustrated in this review article (DOI: 10.7554/eLife.25916, Figure 1 – now Ref. [44] in the revised manuscript). The FIB-SEM is ideal for acquiring high resolution 3D image reconstructions at high resolution – as we have demonstrated here – but for large volume imaging it should be coupled with other high throughput techniques to identify these regions of interest.

Comment

- Is there any conceivable way to put one of these RVEs under mechanical loading?

Authors' response: There is indeed a conceivable way to use these RVEs to study the tissue's mechanical response at the microstructural level. This is what we are currently investigating so that these RVEs can become the input of computational models such as those we have recently presented in Refs. [63-66].

Comment

- Are there any noticeable intra-RVE connections or networks that can be mapped out?

Authors' response: this is indeed a very interesting question. We have recently used both the data as acquired from the FIB-SEM images (Ref. [66]) and our RVEs (work in progress) to study the tissue permeability and, therefore, link the intra-RVE connections (networks) to fluid flow via convection, which is at the basis of the optimisation of CED drug delivery techniques and, we also believe, defines the route for their improvement.

Comment

Thank you for allowing me to review this paper. It is really an impressive work.

Authors' response: Many thanks for reviewing the paper and providing such constructive feedback!

Reviewer #3 (Remarks to the Author):

In this work, the authors study the brain microstructure of the white matter in ovine samples. For that, they used 3D-electron microscopy to reconstruct myelinated fibers in selected areas of the brain: the corpus callosum, coronal radiata and fornix. Using the segmented axons, the authors estimate parameters of interest associated with the axonal morphology.

Comment

1. The authors present in this study “the first systematic attempt...” to characterize the microstructure of white matter. However, many studies have already described the architecture of the white matter using a variety of modalities from traditional 2D light and electron microscopy to advanced 3D light and electron microscopy or x-ray tomography between other methodologies, e.g. polarized light microscopy or coherence tomography. The authors must revise previous studies and acknowledge the evolution of the methodology from traditional to advanced in the context of white matter. In the current version of the manuscript, it is not clear what this study add to the study of white matter.

Authors' response: Many thanks for this very useful suggestion; we have now revised the paper to provide both a more comprehensive review of the advanced made in the past decades to study the architecture of white matter and to highlight the novelty of this study in this context – see also response to point 2 below.

Comment

2. A big limitation in this study is that the authors introduce a manual approach for axonal segmentation, while several studies have been already developed automated and semi-automated segmentation methods for the white matter for higher and lower image resolution as acquire in this work. Few examples of latest published studies on 3D electron microscopy or x-ray tomography offer interesting pipelines for segmentation and quantification of white matter samples:

Along-axon diameter variation and axonal orientation dispersion revealed with 3D electron microscopy: implications for quantifying brain white matter microstructure with histology and diffusion MRI. Lee HH, Yaros K, Veraart J, Pathan JL, Liang FX, Kim SG, Novikov DS, Fieremans E. Brain Struct Funct. 2019 May;224(4):1469-1488. doi: 10.1007/s00429-019-01844-6. Epub 2019 Feb 21.

Automated 3D Axonal Morphometry of White Matter. Abdollahzadeh A, Belevich I, Jokitalo E, Tohka J, Sierra A. Sci Rep. 2019 Apr 15;9(1):6084. doi: 10.1038/s41598-019-42648-2.

Axon morphology is modulated by the local environment and impacts the noninvasive investigation of its structure-function relationship. Andersson M, Kjer HM, Rafael-Patino J, Pacureanu A, Pakkenberg B, Thiran JP, Ptito M, Bech M, Bjorholm Dahl A, Andersen Dahl V, Dyrby TB. Proc Natl Acad Sci U S A. 2020 Dec 29;117(52):33649-33659. doi: 10.1073/pnas.2012533117. Epub 2020 Dec 21.

DeepACSON automated segmentation of white matter in 3D electron microscopy. Abdollahzadeh A, Belevich I, Jokitalo E, Sierra A, Tohka J. Commun Biol. 2021 Feb 10;4(1):179. doi: 10.1038/s42003-021-01699-w.

A semi-automated approach to dense segmentation of 3D white matter electron microscopy. Michiel Kleinnijenhuis, Errin Johnson, Jeroen Mollink, Saad Jbabdi, Karla L. Miller. doi: <https://doi.org/10.1101/2020.03.19.979393>

The authors must review the literature regarding segmentation and quantification of brain samples using 3D methodologies, and demonstrate if their method is more efficient or

consider using a more advanced method. This also affect the quantification: the present work is quite limited in parameters calculated from the axons, while other methods can provide more information of these structures.

Authors' response: Many thanks for this comment. While we were obviously aware of the pertinent literature in this area (for example, paper “Automated 3D Axonal Morphometry of White Matter. Abdollahzadeh A, Belevich I, Jokitalo E, Tohka J, Sierra A. Sci Rep. 2019 Apr 15;9(1):6084. doi: 10.1038/s41598-019-42648-2” was cited as Ref. [61] in our original submission), we thank the reviewer for suggesting some of the most recent literature on the topic. As noted above, the focus of our contribution was on the accurate reconstruction of different parts of the brain and the performance of a comparison between *different regions of the brain* as a conduit to develop accurate microstructural models to be used as inputs for biomechanical investigations. We performed a manual segmentation as this was deemed as the best tool available given the nature and type of data and images acquired to produce the most thorough analysis of the tissues available to us. We do not believe that the specific method used for the segmentation here was to provide a significant element of novelty to the work, therefore we had not indulged in a detailed discussion of the topic in our original submission. Nonetheless, we have now performed an extensive review of this topic and added discussions and some suggestions on the use of more advanced techniques for the reconstruction of 3D microstructures using automated semi-automated and manual methods. We truly believe that our contribution could significantly stimulate the further use of automated and semi-automated techniques, where applicable, to provide an accurate reconstruction of all brain axonal structures within the brain both for the extraction of important parameters and the direct use of reconstructed geometries for modelling purposes. This is now also discussed in the revised manuscript (main text and Methods).

Comment

3. The authors claim on pg 5 that the acquisition protocol was optimized to enable large volumes. $15 \mu\text{m}^3$ is rather small volume in terms of brain microstructural characterization as shown in the papers above.

Authors' response: Many thanks for this remark. Our statement was in the context of optimisation done for the acquisition of multiple volumetric samples from different parts of the brain in a reasonable time (a few hours of FIB-SEM time) while still being able to capture the key microstructural features of the white matter. Thus, “large volumes” was mentioned with this in mind and in relative terms. Note that the same size volume (i.e. $15 \mu\text{m}^3$) was reported as being the standard dimension used to create datasets in Ref. [50] (DeepACSON). We have now re-written this paragraph to avoid potential misunderstandings while also mentioning the possibility of acquiring larger samples as shown by other contributions, now captured in the revised paper.

Comment

4. Are the authors confident that all “white pixels” come from myelin? Are the removed “white pixels” not crucial to estimation of myelin in these data? As seen in Figure 3, there is delamination of the myelin sheaths which can affect the estimation of the diameter of the axons + myelin (what the authors call outer diameter of the axon), estimation of g-ratio (not calculated in this study) or in pathological conditions (such as demyelination). If this is a healthy animal, can the authors explain the delamination of the myelin sheaths?

Authors' response: We do believe that all “white pixels” captured through the image processing procedure described in the Method section of the paper come indeed from myelin. As explained in page 5, in Figure 3d the only operation performed is the elimination of spurious and isolates pixels before background illumination compensation (Figure 3e) and binarization (Figure 3f). if the reviewer refers to the whit pixel removed in the operation leading to Figure 3d, the amount of white pixel identified and removed at this stage is trivial

compared to the amount of white pixel retained and was confined to areas in which are nowhere near the axons' boundaries.

Turning to more interesting and potential more important question of delamination, we do agree with the reviewer that this is something to address. We do observe some limited delamination (although the SEM images in Figure 2 clearly show that this is not a constant features of the images acquired using the methodology shown here), which is not uncommon in EM images of brain matter (see e.g. images in Refs. [48, 52]), and it is usually associated with either tissue damage induced by pathological conditions (to be excluded here) or to alteration induced by the fixation and/or post-mortem time of excision of the brain samples (the latter also being minimised and less of a concern here). We have now discussed this in detail and referred to the relevant literature. We have also mentioned this as a potential limitation of the study and something that should be addressed by standardised protocols to produce high-quality tissue mapping at the microstructural level. It should also be born in mind here that in this study we are particularly interested in the geometrical features concerning the axon+myelin (outer diameter) dimensions as we believe that this is particularly important for the determination of tissue permeability as further investigated in our recent PNAS contribution in Ref. [66]. We believe that the manual segmentation and quality of images obtained here provide a very reasonable set of data to be able to estimate these important parameters, which are in line with similar acquisitions performed in the literature for specific tissue regions, as reported in this article.

Comment

5. The thickness of the slices during imaging was 150 nm. This is a thick slice in terms of microstructural characterization. This should be addressed as a limitation in the discussion.

Authors' response: This is indeed a very good point. As mentioned in the original paper, this slice thickness was chosen to optimise the acquisition time while still being able to study and analyse several samples. In the Methods section we have now both clarified the choice of the thickness, including a mention of the fact that in terms of reconstruction it provides very similar results than those obtained with thinner slices, and explained that this could be a limitation of the methodology, especially in the context of using our images for semi-automatic or automatic segmentation.

Comment

6. Shrinkage with chemical fixation is not negligible as the authors claim on page 16. This has been extensively shown, e.g.: Ultrastructural comparison of dendritic spine morphology preserved with cryo and chemical fixation. Tamada H, Blanc J, Korogod N, Petersen CC, Knott GW. *Elife*. 2020 Dec 4;9:e56384. doi: 10.7554/eLife.56384.

Authors' response: This is certainly an issue we had not intended to disregard or whose importance we had wanted to minimise. In the last part of the discussion section of our original submission (page 16), we wrote: "*Cryo-imaging techniques have also shown to better preserve the ultrastructure of the biological matter when compared to the alcohol-driven dehydration methodology followed in this study,*⁷⁰ *which might have also resulted in limited shrinkage of the sampled tissue.*⁸⁴ *However, the room temperature fixation technique herein adopted, provides a robust, easy and rapid solution for high-throughput imaging of multiple specimens.*" Contrary to the reviewer's interpretation, with this sentence we intended to highlight this as a potential problem (we talk about "limited" rather than "negligible"), which must be considered carefully when samples are prepared for EM studies. This is aligned with the data shown in the paper cited by the reviewer, whereby most measured parameters are in similar Q3-Q1 range when comparing chemical and cryo fixations if they are adequately performed. We apologies if our writing was misleading. Note that we had also cited a previous contribution from the authors of the paper suggested by the reviewer, Ref. [76], and other relevant papers (see e.g. Ref. [104]. Being aware of this issue, we tried to minimise the shrinkage with dehydration as gradual as possible, as now

reported in our revised and improved account of the procedure we used in the Method section. We have also modified our original statement to clearly point out that this as a limitation and to point the reader to the relevant literature (including the paper suggested by the reviewer) to provide a more comprehensive review of the topic.

Comment

7. Because of the shrinkage, the extracellular space is basically gone in chemically fixed samples. Are the authors able to see extracellular space? if so, where? could the authors point at the space in the figures?

Authors' Response: We are not sure we understand this remark as we believe we made it clear, and it was obvious from the images reported in Figure 3, that the extracellular space is characterised by the “black pixels” in our images (that is after binarization) which surround the axons+myelin regions (see Figure 3f for example); it is therefore far from “gone”. The 3D reconstructions in Figure 7 also clearly show gap areas between the 3D axons being reconstructed meticulously. The distance between axons varies from a few to hundreds of nanometers – we have clarified this in the revised text of the manuscript.

Related to that, if the shrinkage minimizes the extracellular space, the reconstructed axons could have very little used to model, for example, drug delivery, as the authors claim in the introduction. This issues must be addressed in the discussion section.

Authors' Response: As mentioned above, the fact that we can indeed distinguish between extracellular matrix and axon+myelin regions allows us to use these model for exactly this purpose, as shown effectively in one of our recent contribution in PNAS (. [66]), which uses this data set as input for detailed models.

Comment

8. The authors must explain in more detail the methodology.

Authors' Response: we agree with the reviewer here that we could have provided more details about the adopted methodology. We have now done so, and we very much hope that the revised version of the manuscript provides all the information needed to address the reviewers' concern in this respect.

Comment

9. The discussion is long, and many points are trivial, e.g. explanation about basic neuroscience.

Authors' Response: Many thanks for this comment. We have tried out best in the revised version of the manuscript to eliminate trivial points while keeping the discussion as broad as possible to cater for the broadest of audiences, who, as the reviewers of this manuscript, may come from different backgrounds. For example, we have removed entire paragraphs from the Discussion and moved some of the relevant information in the Introduction section. We hope that this has improved the flow and the readability of the paper.

REVIEWERS' COMMENTS:

Reviewer #1 (Remarks to the Author):

The authors have thoroughly revised the manuscript and have sufficiently addressed all my comments.

Reviewer #2 (Remarks to the Author):

Thanks for making the changes. great paper!